# Innovative Bioactive Ca-SZ Coating on Titanium Dental Implants: A Multidimensional Structural and Elemental Analysis

**DOI:** 10.3390/jfb15060155

**Published:** 2024-06-05

**Authors:** Alex Tchinda, Aurélien Didelot, Patrick Choquet, Augustin Lerebours, Richard Kouitat-Njiwa, Pierre Bravetti

**Affiliations:** 1Department of Micro and Nanomechanics for Life, Jean Lamour Institute, University of Lorraine, UMR 7198, 54011 Nancy, Francerichard.kouitat@univ-lorraine.fr (R.K.-N.);; 2Materials and Technology Department, Luxembourg Institute of Science and Technology, 41 Rue du Brill, L-4422 Belvaux, Luxembourg

**Keywords:** coating, Ca-SZ, adhesion, biomaterials, implants

## Abstract

The design of new, biomimetic biomaterials is of great strategic interest and is converging for many applications, including in implantology. This study explores a novel approach to improving dental implants. Although endosseous TA6V alloy dental implants are widely used in oral implantology, this material presents significant challenges, notably the prevalence of peri-implantitis. Therefore, in this study, we investigate a new advance in the design of hybrid medical devices. This involves the design of a Ca-SZ coating deposited by PVD on a TA6V substrate. This approach aims to overcome the inherent limitations of each of these materials, namely TA6V’s susceptibility to peri-implantitis on the one hand and zirconia’s excessively high Young’s modulus compared with bone on the other, while benefiting from their respective advantages, such as the ductility of TA6V and the excellent biocompatibility of zirconia, offering relevant prospects for the design of high-performance implantable medical devices. This study integrates characterisation techniques, focusing on the structural and elemental analysis of the Ca-SZ coating by XRD and TEM. The results suggest that this coating combines a tetragonal structure, a uniform morphology with no apparent defects, a clean interface highlighting good adhesion, and a homogeneous composition of calcium, predisposing it to optimal biocompatibility. All of these findings make this innovative coating a particularly suitable candidate for application in dental implantology.

## 1. Introduction

The advent of technological innovation is opening up new possibilities and promising prospects in the science of biomaterials. The design of new, biomimetic biomaterials is of great strategic interest and is converging for many applications, in this case, implantology. Although TA6V alloy endosseous dental implants are widely used in oral implantology, this material presents significant challenges, including the prevalence of peri-implantitis and susceptibility to allergic reactions for some patients [1]. Indeed, the clinical literature reports close correlations between TA6V implants and the onset of allergy in certain patients [2]. The prevalence remains low, at just 0.6% according to a clinical study of 1500 subjects [3]. Other sources report a relatively similar prevalence. However, the method of diagnosis and assessment is the subject of controversy [4]. On the other hand, clinicians are unanimous on the need to contain peri-implantitis, which is a source of great clinical concern. Peri-implantitis is an infectious disease leading to the inflammation of peri-implant tissues, with progressive marginal bone loss associated with soft tissue necrosis around a dental implant, resulting in the loss of an implant through loosening [5]. Several converging aetiological factors are involved, the main one being bacterial in origin. Anaerobic bacteria are capable of colonising TA6V implant surfaces, leading to corrosion of the implant by the acidification of the surface as a result of their metabolic activity [6,7]. Given that TA6V alloy dental implants are composed of aluminium and vanadium, it is clear from an in vivo clinical study that vanadium and aluminium from TA6V implants are released into the systemic bloodstream following corrosion of the implants by bacterial biofilm [8]. Vanadium has a cytotoxic effect, while aluminium can cause peri-implant osteolysis, as well as local inflammation and significant neurotoxic effects, according to Sridhar et al. [9]. A recent systematic review of a meta-analysis of data reported a weighted prevalence of 9.25% for peri-implantitis, suggesting an increasing observation of this condition over time [10].

Although the allergenic potential of TA6V is considered low, it nevertheless represents an obstacle for some patients who prefer alternatives. Even more worrying is the prevalence of TA6V in peri-implantitis, which is a major source of concern for clinicians, calling its use into question. Taken together, these observations translate into converging risk factors, prompting the emergence of the concept of ‘metal-free’ in oral implant dentistry. This concept is based on the use of materials that are free from the aforementioned metal components and materials other than TA6V in the manufacture of dental implants, with yttria zirconia (Y-TZP) implants as a concrete example. This is a zirconia stabilised by the addition of 3–4 mol% yttrium oxide (Y_2_O_3_) and a tetragonal crystallographic phase, giving it greater tensile and compressive strength than TA6V and making it particularly suitable for oral implantology [11]. Better still, the literature reports a substantial body of information ranging from preclinical studies to clinical studies, mentioning the excellent biocompatibility of Y-TZP in contact with human gingival fibroblasts, according to Magne et al. [12]. The low inflammatory potential was demonstrated by Djidi et al. to be at the junction between a zirconia implant and the cells of the peri-implant mucosa, following a randomised prospective histological and immunohistochemical study [13]. Similarly, an in vivo osseointegration study in rabbits showed that the contact surfaces between bone and zirconia implants were greater than the contact surfaces between bone and TA6V implants after 12 weeks of bone healing [14]. More importantly, an in vitro and in vivo study on bacterial colonisation suggests that zirconia implant surfaces do not favour bacterial colonisation, preventing bacterial biofilm formation and therefore peri-implantitis, which makes this material a preferred alternative to TA6V [15]. These favourable observations on the biological advantages of zirconia are far from exhaustive, as is the aesthetic satisfaction derived from its white colour, which is reminiscent of a natural tooth.

In this study, we examine an innovative advance in the design of hybrid medical devices. This involves the design of a Ca-SZ coating deposited by PVD on a TA6V substrate, in order to exploit one of the advantageous mechanical properties of TA6V, such as ductility, and to benefit from the excellent biocompatibility of zirconia compared with TA6V. This approach aims to overcome the inherent limitations of each of these materials, namely TA6V’s predisposition to peri-implantitis on the one hand and zirconia’s excessively high Young’s modulus compared with bone on the other, while benefiting from their respective advantages, such as TA6V’s ductility and zirconia’s excellent biocompatibility, offering relevant prospects for the design of high-performance implantable medical devices. Doping Ca-SZ coatings with heterovalent ions of lower valency than zirconium, such as calcium oxide, as a replacement for yttrium oxide, represents a strategic approach to improving the biocompatibility and osseointegration of dental implants. This innovation aims to limit bacterial colonisation leading to peri-implantitis due to the presence of zirconia on the surface while promoting positive osteogenic responses of peri-implant bone tissue by bioactivation, in particular by the chemotaxis-directed stimulation of osteoclasts, which play an important role in the process of bone healing and reconstruction by resorption of calcium particles at the bone–implant interface [15,16]. Thus, this study integrates advanced multidimensional characterisation techniques, focusing on the structural and elemental analysis of the Ca-SZ coating to provide an in-depth picture of the influence of calcium doping on the crystalline structure of Ca-SZ and its impact on the internal microstructure of this innovative coating.

## 2. Material and Methods

### 2.1. Reactive Sputter Deposition of Ca-SZ

The Ca-SZ coatings were deposited by reactive magnetron sputtering onto ø 2 cm, 2 mm thick TA6V alloy substrates supplied by Visy implant^®^ (Chavanod, France). The substrates were ultrasonically cleaned in ethanol and then air-dried. Deposits were made in a vacuum chamber equipped with two 2-inch diameter cathodes, one made of zirconium at 99.99% purity and an applied electrical power of 0.45 A, and the other made of calcium at 99.99% purity and an applied electrical power of 0.10 A. The initial vacuum was of the order of magnitude of 10^−7^ mbar. The working pressure was maintained at 1 Pa, and then an argon flow rate of 30 sccm was introduced, and an oxygen flow rate of 10 sccm was also introduced into the EVA300 deposition chamber (Institut Jean Lamour, Nancy, France). A rolling valve, monitored by a computer, was used. The electrical power applied to the cathodes was regulated by two pulsed DC generators with a frequency of 100 kHz and a rest time of 2 µs for the Zr cathode and 5 µs for the Ca cathode. The deposition rate was measured at 2.08 nm/min. The target-substrate distance was 6 cm, and the sample holder was rotated at 6 rpm to ensure spatial homogeneity of the deposits. Thicknesses of 250 nm, 450 nm, and 850 nm were obtained by varying the deposition time. The 850 nm thickness was selected for this study.

### 2.2. Phase Identification by X-ray Diffraction

The crystalline phase of the coating was identified using a Bruker^®^ D8 Advance AXS GmbH (Karlsruhe, Germany) diffractometer, equipped with a copper-anode X-ray tube and a Lynxeye detector. The radiation used was Cu Kα 1, with a wavelength of 1.5406 Å. Measurements were performed in θ-2θ mode. The angular range explored was 10° to 110° in 2θ. Diffrac.Eva V6.0.07 (64-bit) software was used to analyse the DRX spectra for phase identification based on the ICDD PDF-2 2023 database.

### 2.3. Lamella Extraction by FIB

A Helios NanoLab 600i SEM-FIB (Institut Jean Lamour, Nancy, France) was used to take a thin slice from the Ca-SZ coating deposited by PVD on the TA6V substrate. The area of interest was protected by a 15 × 2 × 2 µm^3^ platinum deposit. The contours were carefully cut. Next, a 1.5 µm thick slide was extracted, transferred for thinning at 30 kV, and then refined at low voltage at 5 kV-15 pA, then at 2 kV, and finally at 1 kV. The slide was then observed in SE mode at 5.00 kV.

### 2.4. Transmission Electron Microscopy

The FIB lamella was analysed by corrected transmission and scanning electron microscopy (TEM/STEM) on a JEOL JEM-ARM 200F Cold FEG (Institut Jean Lamour, Nancy, France), dedicated to high-resolution structural and chemical analysis. This microscope operates at 200 kV and offers a resolution of 0.08 nm in STEM mode and 0.19 nm in TEM mode. The images obtained were analysed using DigitalMicrograph^®^ GATAN version 3.40.2804.0 software.

## 3. Results

### 3.1. Phase Identification by X-ray Diffraction

Figure 1 shows the X-ray diffraction pattern of the bare TA6V substrate coated with 850 nm of PVD-deposited Ca-SZ. The coating reveals a tetragonal zirconia crystal phase, identified by PDF 04-024-5383. This phase has a tetragonal lattice of space group P42/nmc (No. 137) and lattice parameters a = 3.60 Å and c = 5.18 Å. The TA6V substrate has a hexagonal organisation with lattice parameters a = 2.93100 Å and c = 4.66900 Å for a space group of p63/nmc (194).

The diffraction peaks characteristic of the tetragonal phase of the coating are clearly visible in the enlarged areas of Figure 1. They are absent from the TA6V substrate in black but present for the Ca-SZ in blue. These correspond to the crystal planes (101) around 30°, (002) and (110) around 35°, (112) and (200) around 50°, and (103) and (211) around 60°, respectively.

Figure 2 compares the diffraction patterns of Ca-SZ coatings of different thicknesses: 250 nm in black line, 450 nm in blue line, and 850 nm in red line. It should be noted that the crystalline phase remains unchanged, but there is a slight tendency for the width of the peaks to increase with the thickness of the coating around 35°, 50°, and 60° (area b and c), unlike 30°, which remains constant (area a).

### 3.2. Micrographs of FIB Lamella

SEM micrographs reveal distinct aspects of the FIB lamina, with the top third showing a light hue, consisting of the Ca-SZ coating and a top protective platinum layer deposited during extraction (Figure 3). In contrast, the lower third has a slightly darker hue, constituting the TA6V substrate. The interface between these two materials is clearly distinguishable, demonstrating a continuous interface with no structural gaps. However, the vertical raisings visible on the lamella, and pronounced in the substrate area, are caused by the beam cut during the lamella thinning step; moreover, they are oriented according to the angle of the beam.

### 3.3. Transmission Electron Microscopy Analysis

#### 3.3.1. TEM and HR-TEM Micrographs

The TEM micrograph of the cross-section of the Ca-SZ/TA6V coating at different magnifications reveals the characteristics of a coating adherent to the substrate (Figure 4). Indeed, the absence of gaps and structural defects in the junction zone reveals a clean interface with no delamination. Although a thin amorphous layer is present at the transition interface, the coating has a fine filamentous microstructure perpendicular to the substrate with a homogeneously organised microstructure. It should be noted that the presence of vertical striations is the result of the thinning of the lamella by FIB.

Even more accurately, HR-TEM micrographs of the Ca-SZ coating at different magnifications show variations in contrast and texture through characteristic shades of grey (Figure 5a,c). The overall compound patterns reflect a certain degree of crystallite and atomic organisation within the material involving randomly oriented lattices. However, this random orientation and the absence of well-defined and regular atomic planes suggest the presence of several crystalline phases in the coating. On the other hand, some spots appear highlighted, indicating an orientation of the crystals in the [002] direction (Figure 5d). On the other hand, the image-simulated diffraction patterns using the fast Fourier transform (FFT) are sharp, intense, and arranged in concentric rings (Figure 5b,d), showing differentially oriented hkl lattice planes and confirming the polycrystalline nature of the Ca-SZ coating. In contrast, the regular single-crystal grating pattern of TA6V can be seen in the simulated zone diffraction pattern in Figure 6 for comparison.

Although the orientation of the hkl lattice planes of the substrate is random, it should be noted that the hkl crystal lattice planes obtained by the fast Fourier transform (FFT) model correspond to the hkl obtained by DRX above. This concordance of data confirms the crystalline nature of this material.

#### 3.3.2. STEM Micrography

The HAADF (high-angle annular dark field) detection modes in Figure 7a,c,e and the BF (bright field) mode in Figure 7b,d,f at different magnifications of the Ca-SZ/TA6V interface show a homogeneity of the different chemical contrasts in superimposed images, suggesting a uniform distribution of the chemical elements making up the coating. Nevertheless, the presence of vertical abrasion striations running from the substrate to the coating resulting from the thinning process of the lamella by FIB remains very visible. These striations reveal hollow, light, and dark areas on either side that should not be confused with the morphology of the coating.

The sensitivity of the HAADF mode to variations in atomic number enables us to observe variations in electron density and therefore in chemical composition on a nanometric scale, with a marked contrast appearing between the Ca-SZ coating (lighter) and the TA6V substrate (darker). In contrast, in the BF mode, the contrast is less pronounced, although the details remain clearly visible. Here, the filamentous appearance of the coating described above suggests a lamellar-type structure. In addition, the interface is sharp and distinct with no noticeable inter-scattering, forming a black-and-white line in both HAADF and BF modes that is clearly distinguishable at high magnification. Most importantly, there are no junction defects such as cracks, blisters, pores, or inclusions in the interfacial area.

#### 3.3.3. Mapping and EDS Analysis

There appears to be a homogeneous distribution of each chemical element and a circumscribed localisation of oxygen, zirconium and calcium in the Ca-SZ coating, and titanium and aluminium in the TA6V substrate, despite some images showing diffuse noise (Figure 8). However, the absence of vanadium, which was not obtained due to the proximity of the Kβ lines of titanium, and the Kα lines of vanadium, which are too close in EDS to be distinguished, should be noted. The semi-quantitative elemental composition along the transition interface from Figure 9 shows a differential evolution of the chemical elements present in the substrate and the coating (Table 1). It should be noted that the “LG10006” transition point is the point at the interface between the coating and the substrate at which the general elemental composition reverses (Figure 10). Thus, the coating was found to be composed of approximately 63% oxygen, 31% zirconium, and 5% calcium at the top of the “LG10014” profile (Figure 10), which is relatively close to the chemical formula obtained by XRD. However, it is worth noting the presence of minute traces of titanium (less than 1%) in the coating, which can be explained by contamination due to re-deposition during the FIB thinning process, given the hardness and majority composition of titanium in the TA6V substrate.

Table 1 shows X-ray maps of the Ca-SZ/TA6V cross-section at 1000 k magnification using STEM micrographs in the HAADF mode.

This shows a much clearer and more precise elemental distribution, as well as an absence of inter-diffusion of chemical elements across the interface (Figure 11). As described previously, the semi-quantitative elemental composition along the transition interface from Figure 12 shows a differential elemental evolution (Table 2). Here, the breakpoint is “LG20004”, although the high magnification makes it possible to accurately observe an inversion at the start of the profile. However, even at high magnification at the interface, there is no significant change in the semi-quantitative composition of the Ca-SZ coating (Figure 13).

## 4. Discussion

The various characterisation methods used in this study enable several levels of observation and understanding of the properties and potential applications of the Ca-SZ coating. The XRD analyses of the Ca-SZ coating produced by PVD reveal the presence of a tetragonal-type crystalline structure, marked by the presence of the characteristic peaks observed in Figure 1. The DRX analysis shows that the width of the peaks in the diffractogram in Figure 2 increases slightly with the thickness of the coating around 35°, 50° and 60°, whereas 30° remains constant. On the face of it, the broadening would probably be due to a significant electronic interaction as a function of the amount of material traversed by the X-rays. In contrast, the constancy of the peak at 30° corresponds to an interplanar distance that does not seem to be affected by this factor.

However, HR-TEM analyses show a polycrystalline coating with differential orientation and a homogeneous filamentous structure with no apparent structural defects.

Elsewhere, TEM, HR-TEM, and STEM micrographs reveal an absence of junction defects such as cracks, blisters, pores, or inclusions in the interfacial zone. However, the presence of a thin, amorphous interfacial line should be emphasised, which could explain this observation. It should be noted that this thin line may be a layer of oxide on the surface of the substrate. Indeed, the literature reports that TA6V alloys, due to the phenomenon of passivation of titanium, have a permanent oxide layer on the surface [17,18,19].

STEM analyses reveal a uniform distribution of the chemical elements making up the coating, which is confirmed by progressive magnification EDS mapping. The semi-quantitative elemental composition from the points along the interface highlights the dominant presence of oxygen and an absence of elemental interdiffusion. This is one of the objectives of the coating, which is to confine the diffusion of toxic aluminium, vanadium, and titanium particles caused by implant corrosion into the systemic circulation [20,21]. However, the homogeneous distribution of approximately 4% calcium in the coating would play an important role in the stabilisation of the tetragonal phase, as well as in the process of cellular bioactivation to initiate osseointegration and bone healing [22,23].

As the Ca-SZ coating deposited by PVD provides valuable information, it is appropriate to refocus the various perspectives around the contribution of these promising observations to innovation. In addition to our innovative objectives of promoting bioactivity by functionalising surfaces to optimise osseointegration of dental implants, this coating could find relevant applications both for implantable medical devices in general and in industry. However, it should be pointed out that additional avenues of exploration are needed, such as relaxation annealing to observe the evolution of the crystalline phase. Exploring Rietveld refinement to quantify the different crystalline phases present in the material would also be of interest. It would also be relevant to explore the surface oxide layer using Raman spectroscopy, as well as EBSD, which would enable phase mapping and the measurement of orientation, recrystallisation, and grain growth.

## 5. Conclusions

The Ca-SZ coating produced by PVD on TA6V substrates has very encouraging characteristics, making this innovative coating a particularly suitable candidate for application in dental implantology. This coating combines a tetragonal structure, a uniform lamellar morphology with no apparent defects, a clean interface highlighting good adhesion, and a homogeneous composition, as well as an absence of elemental interdiffusion. These qualities predispose the Ca-SZ coating to optimal biocompatibility. That, however, must be validated by in vivo studies. Clinical studies need to be conducted to validate the long-term results as well as the potential advantages of this coating in preventing peri-implantitis.

## Figures and Tables

**Figure 1 jfb-15-00155-f001:**
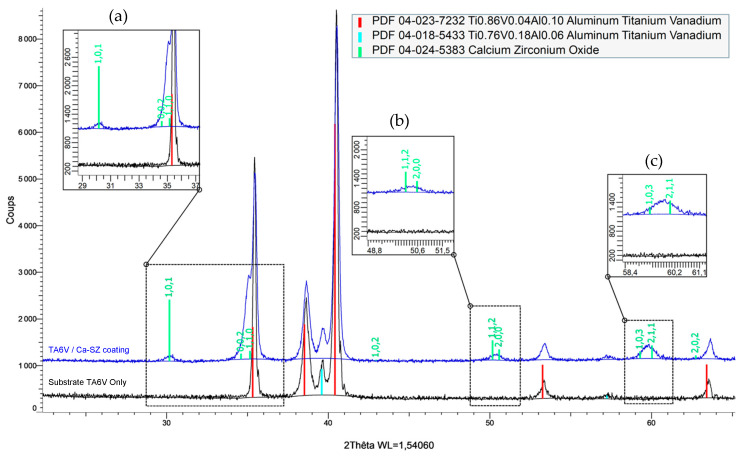
Diffraction pattern of the 850 nm thick Ca-SZ coating deposited by PVD on a TA6V substrate.

**Figure 2 jfb-15-00155-f002:**
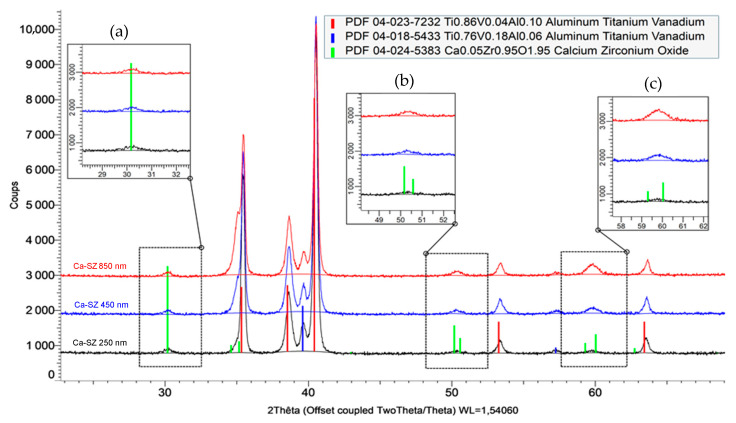
Comparative diffraction pattern of 250, 450, and 850 nm thick Ca-SZ coatings deposited by PVD on a TA6V.

**Figure 3 jfb-15-00155-f003:**
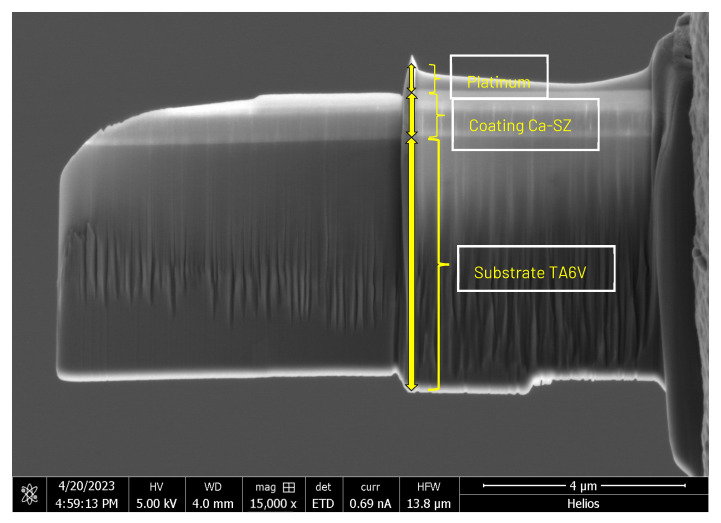
SEM image of the FIB lamella obtained from an 850 nm thick coating of Ca-SZ deposited by PVD on a TA6V substrate (magnification ×15,000 SEM FIB Helios NanoLab 600i).

**Figure 4 jfb-15-00155-f004:**
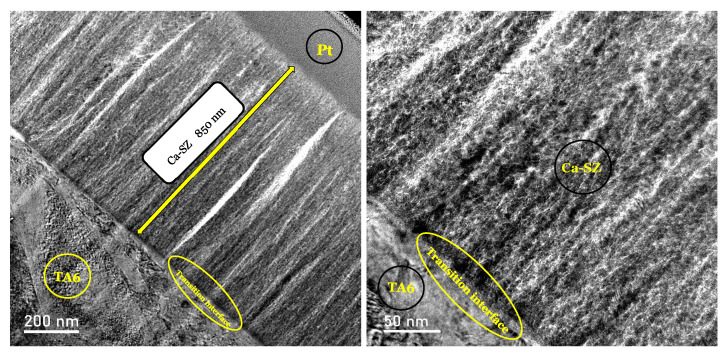
TEM micrograph of the transverse subsurface of the Ca-SZ coating (indicated by the yellow arrow), (× magnification 11,877.1 on the left and 48,898.2 on the right) (transition interface indicated by a circle).

**Figure 5 jfb-15-00155-f005:**
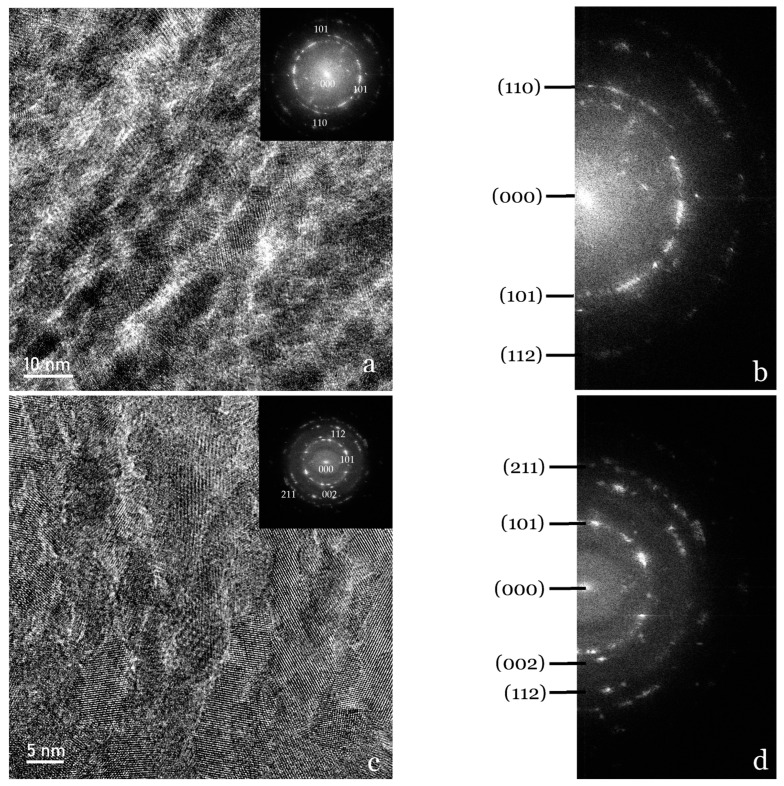
HR-TEM micrographs of the Ca-SZ coating at various magnifications, accompanied by a fast Fourier transform insertion (**a**,**c**). The hkl planes of the corresponding grating are identified in the simulated diffraction patterns on images (**b**,**d**).

**Figure 6 jfb-15-00155-f006:**
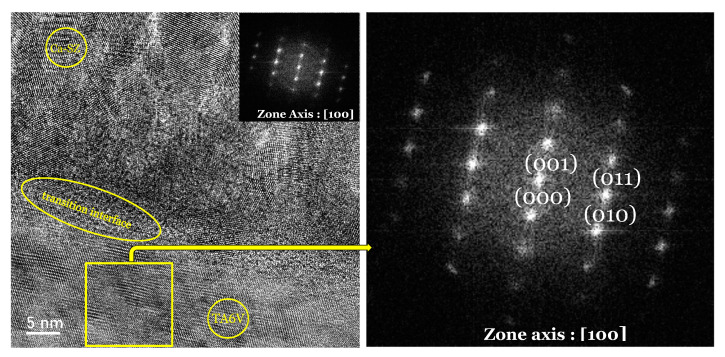
HR-TEM micrograph of the TA6V/Ca-SZ interface with fast Fourier transform insertion. The hkl planes of the corresponding grating are identified on the simulated diffraction images of the TA6V zone (square frame indicated by the arrow). The encircled zone between the Ca-SZ coating and the TA6V substrate is the interfacial trasnistion line.

**Figure 7 jfb-15-00155-f007:**
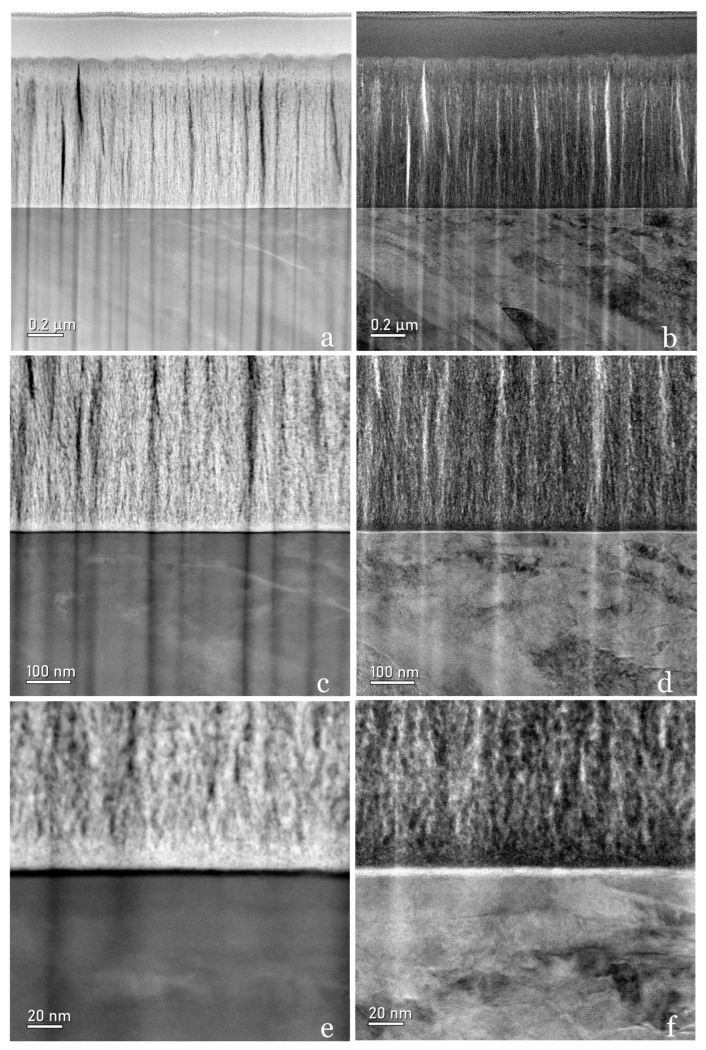
STEM micrograph at progressive magnification. On the **left**, dark field annular imaging sensitive to chemical contrast (HAADF) is shown, and on the **right**, bright field (BF) annular imaging sensitive to light elements.

**Figure 8 jfb-15-00155-f008:**
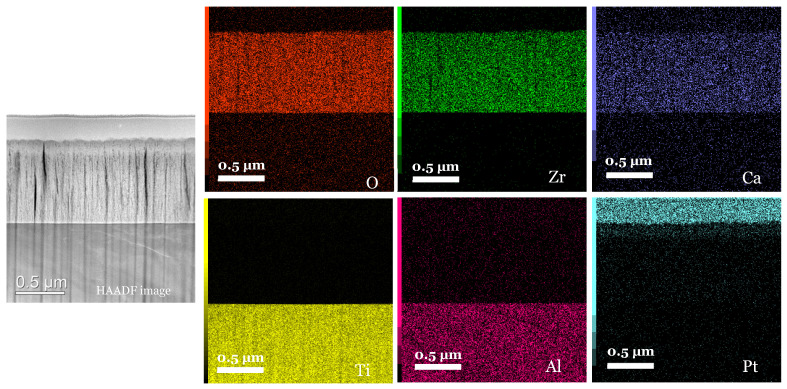
X-ray maps of the Ca-SZ/TA6V interface (magnification 100 k).

**Figure 9 jfb-15-00155-f009:**
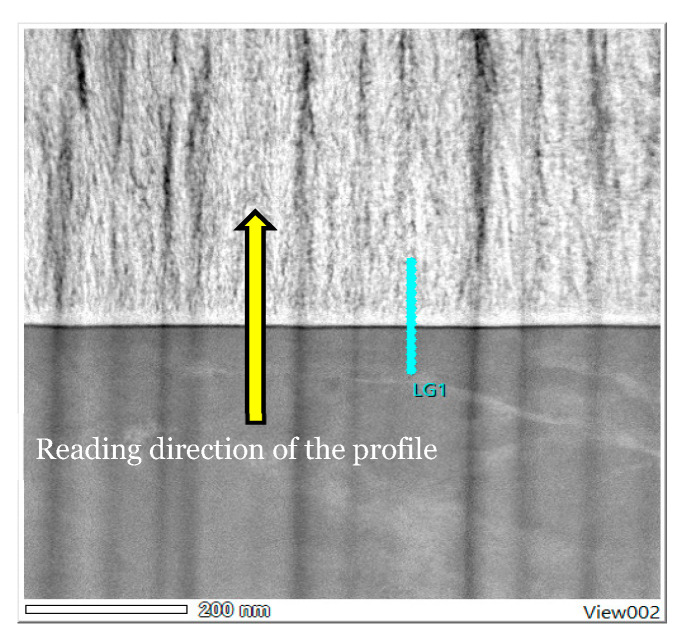
HAADF EDS point image (magnification 100 k).

**Figure 10 jfb-15-00155-f010:**
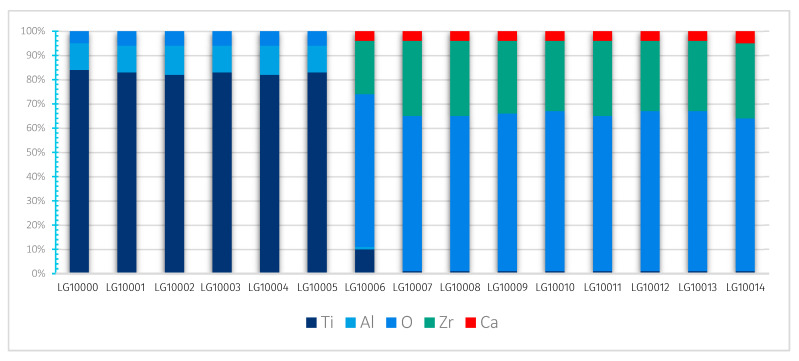
Evolution profile and semi-quantitative composition (magnification 100 k).

**Figure 11 jfb-15-00155-f011:**
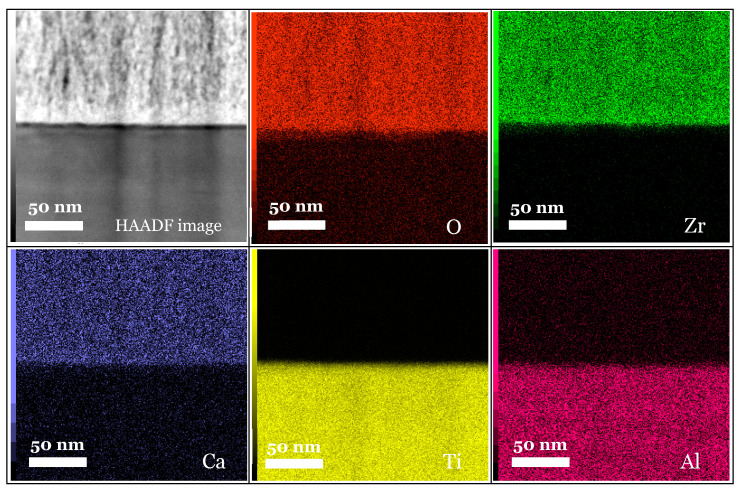
X-ray maps of the Ca-SZ/TA6V interface (magnification 1000 k).

**Figure 12 jfb-15-00155-f012:**
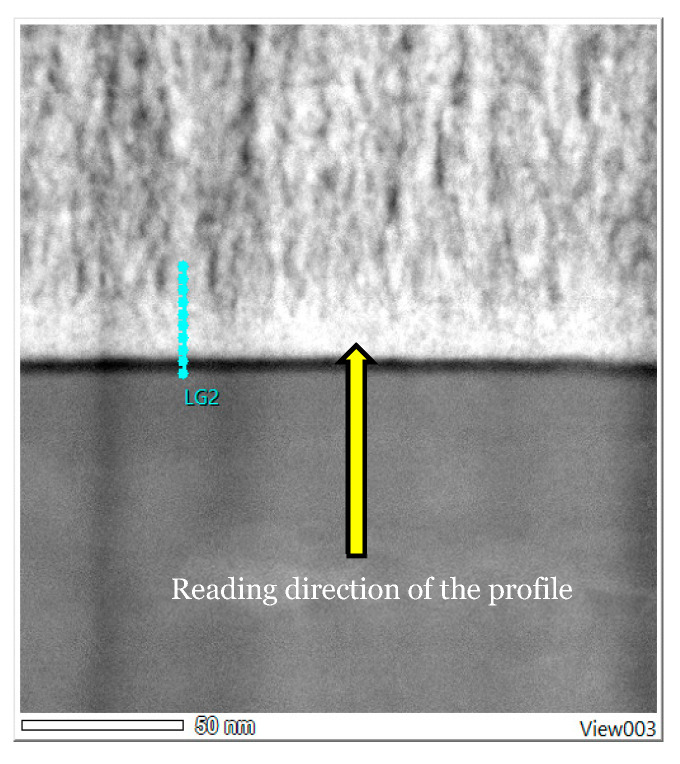
HAADF EDS point image (magnification 1000 k).

**Figure 13 jfb-15-00155-f013:**
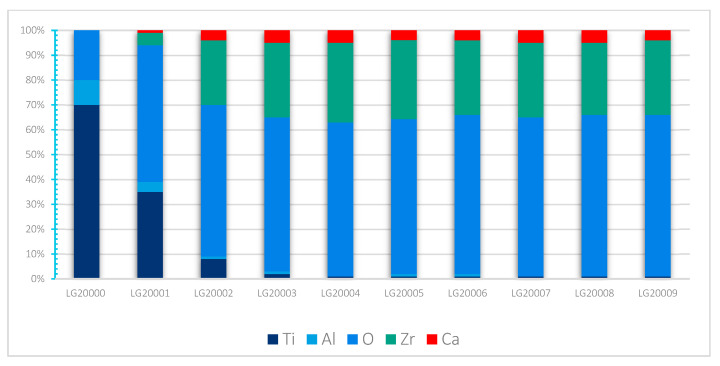
Evolution profile and semi-quantitative composition (magnification 1000 k).

**Table 1 jfb-15-00155-t001:** Semi-quantitative EDS data points (magnification 100 k).

Profile	O	Al	Ca	Ti	Zr	Total (Atom%)
LG10000	5	11	0	84	0	100
LG10001	6	11	0	83	0	100
LG10002	6	12	0	82	0	100
LG10003	6	11	0	83	0	100
LG10004	6	12	0	82	0	100
LG10005	6	11	0	83	0	100
LG10006	63	1	4	10	22	100
LG10007	64	0	4	1	31	100
LG10008	64	0	4	1	31	100
LG10009	65	0	4	1	30	100
LG10010	66	0	4	1	29	100
LG10011	64	0	4	1	31	100
LG10012	66	0	4	1	29	100
LG10013	66	0	4	1	29	100
LG10014	63	0	5	1	31	100

**Table 2 jfb-15-00155-t002:** Semi-quantitative EDS data (magnification 1000 k).

Profile	O	Al	Ca	Ti	Zr	Total (Atom%)
LG20000	20	10	0	70	0	100
LG20001	55	4	1	35	5	100
LG20002	61	1	4	8	26	100
LG20003	62	1	5	2	30	100
LG20004	62	0	5	1	32	100
LG20005	63	1	4	1	32	100
LG20006	64	1	4	1	30	100
LG20007	64	0	5	1	30	100
LG20008	65	0	5	1	29	100
LG20009	65	0	4	1	30	100

## Data Availability

Data are contained within the article.

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
