# Peer review of "Innovative Bioactive Ca-SZ Coating on Titanium Dental Implants: A Multidimensional Structural and Elemental Analysis"

_jfb, 2024, doi:10.3390/jfb15060155_

Round 1

Reviewer 1 Report

Comments and Suggestions for Authors

This study investigates a novel approach to enhancing dental implants by designing a Ca-SZ coating deposited on a TA6V substrate using PVD. The goal is to address challenges associated with TA6V alloy implants, such as peri-implantitis, by combining the advantages of both materials. The Ca-SZ coating aims to overcome limitations of TA6V's susceptibility to peri-implantitis and zirconia's high Young's modulus compared to bone. The study utilizes characterization techniques, including XRD and TEM, to analyze the coating's structure and elemental composition. Results suggest the coating has a tetragonal structure, uniform morphology with good adhesion, and a homogeneous composition of calcium, indicating optimal biocompatibility. Overall, the paper presents an interest to the scientific community. However, some points need clarification. The use of images is not adequate - lack of information and readability. The discussion is relatively short and the data are mainly observational. The authors could use their data better to imporve the strength of their findings. Here are the following comments that should be adressed : 

Majors comments : 

In Figure 2, the authors examined the diffraction patterns of Ca-SZ coatings with various thicknesses (156 nm, 250 nm, 450 nm, and 850 nm). However, the legends indicate different types of coating. Could you please provide clarification on this matter?

It appears that the homogeneity of their coating is crucial. Please provide further details on this and describe the parameters used to address this issue, such as the absence of junction defects. Discuss this aspect in relation to the scales used and the technical limitations.

The semi-quantitative elemental composition from the points along the interface highlights the predominance of oxygen and the absence of elemental interdiffusion. I assume this is related to the transition zone. Could you elaborate on it further?

The authors stated that the 850 nm thickness was chosen for the study. Why were the 250 nm and 450 nm coatings also investigated? What are the advantages and disadvantages of each thickness?

Although the authors claimed that the crystalline phase remains unchanged, they noted a change in the peak width. What about the peak-to-width ratio at half peak? It would be beneficial for the authors to provide more data and analyze their graphs rather than solely presenting them.

The DRX analysis reveals that the width of the peaks in the diffractogram in Figure 2 slightly increases with the thickness of the coating at around 35°, 50°, and 60°, while 30° remains constant. Please provide a quantitative evaluation of this observation.

Minor comments : 

Figures 1 and 2 are somewhat confusing in terms of what outcomes should be extracted from them. The authors should offer clear outcomes, specify which parameters are significant, articulate their working hypothesis, and identify the primary outcomes demonstrating the success of their intervention.

Regarding Figure 3, please provide more details on the location of the coating. Simply displaying the image without a clear legend is not helpful.

In Figure 4, since the authors circled the transition interface, please also zoom in on the TA6V, Ca-SZ, and PT to illustrate the differences better. Despite the differences in scale, please explain what readers should understand from the image. For example, is the thin amorphous layer present at the transition interface?

For Figure 5, where the hkl planes of the corresponding grating are identified in the simulated diffraction patterns on images (b) and (d), please elaborate on this and clarify what readers should comprehend from it.

Figure 6 contains yellow circles and ellipses that are not described. Could you please clarify the intended message? Furthermore, the authors should describe the hkl planes images they are presenting. Are they related to axes? This should be clarified in the legend. Additionally, elaborate on the qualitative evaluation of these images.

Figures 7 and 8 require elaboration on the legend and what readers should understand from them.

Figure 9 : please provide more information on the transition LG100006 and LG 100005.

In Figure 10, the authors show LG2 on the figure. What is this related to? Are there 9 measurements? The same applies to Figure 11.

Comments on the Quality of English Language

ok

Author Response

Reviewer 1:

Comments and Suggestions for Authors

This study investigates a novel approach to enhancing dental implants by designing a Ca-SZ coating deposited on a TA6V substrate using PVD. The goal is to address challenges associated with TA6V alloy implants, such as peri-implantitis, by combining the advantages of both materials. The Ca-SZ coating aims to overcome limitations of TA6V's susceptibility to peri-implantitis and zirconia's high Young's modulus compared to bone. The study utilizes characterization techniques, including XRD and TEM, to analyze the coating's structure and elemental composition. Results suggest the coating has a tetragonal structure, uniform morphology with good adhesion, and a homogeneous composition of calcium, indicating optimal biocompatibility. Overall, the paper presents an interest to the scientific community. However, some points need clarification. The use of images is not adequate - lack of information and readability. The discussion is relatively short and the data are mainly observational. The authors could use their data better to imporve the strength of their findings. Here are the following comments that should be adressed : 

Majors comments: 

Comment 1: In Figure 2, the authors examined the diffraction patterns of Ca-SZ coatings with various thicknesses (156 nm, 250 nm, 450 nm, and 850 nm). However, the legends indicate different types of coating. Could you please provide clarification on this matter?

 Answer 1: For clarification, this is a single synthesis of a unique Ca-SZ coating deposited by PVD under the same conditions but with variable thicknesses of 250 nm, 450 nm, and 850 nm. Deposition times were adjusted to achieve these different thicknesses, as described in the Materials and Methods section. The objective is to explore not only the intrinsic properties of the coating but also the impact of its thickness on aspects such as modification of the crystallographic phase and mechanical behaviors under stress. The latter are the subject of another complementary study currently being published.

Comment 2: It appears that the homogeneity of their coating is crucial. Please provide further details on this and describe the parameters used to address this issue, such as the absence of junction defects. Discuss this aspect in relation to the scales used and the technical limitations.

 Answer 2: The clearest explanation lies not only in the PVD deposition technique, known for producing composites with remarkable homogeneity and extreme precision, but also in the EDS mapping technique using the semi-quantitative HAADF mode in TEM, which highlights the distribution of chemical elements at both low and high magnification, as presented and explained in the results. This method is particularly complementary to the STEM mode, as it provides valuable information on chemical contrasts in both HAADF and BF modes simultaneously, allowing for the detection of even the slightest defects or the absence of defects at the coating-substrate interface. Indeed, these explanations are detailed in the results section.

Comment 3: The semi-quantitative elemental composition from the points along the interface highlights the predominance of oxygen and the absence of elemental interdiffusion. I assume this is related to the transition zone. Could you elaborate on it further?

 Answer 3: Yes, indeed, the aim here is to highlight the absence of inter-element diffusion along the coating/substrate interface to confirm or refute the "confinement" effect of the coating concerning elements such as titanium and aluminum present in the substrate. We conducted EDS quantification points along the transition interface, both at low magnification for an overall large-scale view and at high magnification for a more detailed precision of the evolution of the chemical elemental composition along the transition interface.

Comment 4: The authors stated that the 850 nm thickness was chosen for the study. Why were the 250 nm and 450 nm coatings also investigated? What are the advantages and disadvantages of each thickness?

 Answer 4: We refer you to the response to comment number 1 for the first part of this question. Regarding the advantages and disadvantages of each thickness, it is a strategic choice focused on the mechanical adhesion of different coating thicknesses on the substrates following significant mechanical stress. This topic is the subject of another ongoing study, which aims to determine the minimum coating thickness that can withstand significant mechanical stress without delamination. Thus, the other mentioned thicknesses were compared solely on a crystallographic basis to demonstrate the preservation of crystalline architecture regardless of thickness, as explained in comment number 1.

Comment 5: Although the authors claimed that the crystalline phase remains unchanged, they noted a change in the peak width. What about the peak-to-width ratio at half peak? It would be beneficial for the authors to provide more data and analyze their graphs rather than solely presenting them.

 Answer 5: As explained in the discussion section, the slight increase in peak width is a well-known phenomenon resulting from increased electron interaction of the diffraction beam with the increasing thickness of the traversed material. Therefore, the mildness of this aspect, which we termed as a "trend," renders a thorough analysis less relevant in this regard. However, we chose to present this aspect for clarity purposes to avoid any potential confusion for the readers.

Comment 6: The DRX analysis reveals that the width of the peaks in the diffractogram in Figure 2 slightly increases with the thickness of the coating at around 35°, 50°, and 60°, while 30° remains constant. Please provide a quantitative evaluation of this observation.

 Answer 6: The response to comment number 5 fully addresses this comment

Minor comments: 

Comment 7: Figures 1 and 2 are somewhat confusing in terms of what outcomes should be extracted from them. The authors should offer clear outcomes, specify which parameters are significant, articulate their working hypothesis, and identify the primary outcomes demonstrating the success of their intervention.

 Answer 7: We have presented the results in a manner that makes them as clear and understandable as possible, in two separate figures to simplify comprehension. Additionally, the indexing of corresponding significant peaks in both figures is clearly visible on the x-axis of Figures 1 and 2. Furthermore, we have clearly articulated our working hypotheses at the end of the introduction section and in the discussion section in light of all the results to demonstrate the success of our interventions.

Comment 8: Regarding Figure 3, please provide more details on the location of the coating. Simply displaying the image without a clear legend is not helpful.

 Answer 8: We apologize for this oversight. Unfortunately, we have noticed that the file you received did not include the various annotations for Figure 3, as well as for other figures. This is due to information loss during the conversion of the original manuscript Word file to the editor format that was sent to you. We have added the missing annotations.

Comment 9: In Figure 4, since the authors circled the transition interface, please also zoom in on the TA6V, Ca-SZ, and PT to illustrate the differences better. Despite the differences in scale, please explain what readers should understand from the image. For example, is the thin amorphous layer present at the transition interface?

 Answer 9: The issue is similar to that of the previous comment. We have added the missing explanatory annotations to the various figures.

Comment 10: For Figure 5, where the hkl planes of the corresponding grating are identified in the simulated diffraction patterns on images (b) and (d), please elaborate on this and clarify what readers should comprehend from it.

 Answer 10: In Figure 5, we elaborated on the arrangement of the hkl planes simply based on contrasting elements from the adjacent Figure 6 to simplify understanding. This can be found in lines 188 to 203.

Comment 11: Figure 6 contains yellow circles and ellipses that are not described. Could you please clarify the intended message? Furthermore, the authors should describe the hkl planes images they are presenting. Are they related to axes? This should be clarified in the legend. Additionally, elaborate on the qualitative evaluation of these images. elaboration on the legend and what readers should understand from them.

 Answer 11: We have added circle annotations directly onto the image, although this was already described in the legend, as well as the square indicating that the hkl planes were developed from a simulated zone diffraction pattern, with the zone axis clearly visible. This figure simply serves as an indicator of the difference in orientation of the crystalline planes between the coating and the substrate. In this regard, providing a qualitative explanation of the substrate was not among the objectives of this study.

Comment 12: Figure 9 : please provide more information on the transition LG100006 and LG 100005.

 Answer 12: As explained in lines 243-251, these points are located in the interfacial zone between the coating and the substrate, where an inversion of the interfacial chemical composition is observed, as explained in response to comment number 3. Further information is provided in the mapping section and in the discussion.

Comment 13: In Figure 10, the authors show LG2 on the figure. What is this related to? Are there 9 measurements? The same applies to Figure 11.

Answer 13: Figure 10 corresponds to the TEM image in HAADF mode from which the semi-quantitative EDS points in Figure 11 were obtained.

Comment 14: Comments on the Quality of English Language : ok

Answer 14: Thank you

Reviewer 2 Report

Comments and Suggestions for Authors

First of all , authors used focused ion beam (FIB), which is indeed a state if the art technique and their results are quite impressive. 
1. Abstract is well written

2.  Regarding related studies only some references are provided RefS. 13-14. It is needed to provide literature corespond to previously conducted studies. 
3. In introduction section some refrences are too old. Additionaly, gap of study is missing which can be tackle after dealing with point 2
4. Experimental portion is well organized
5. Rewrite sentences 157-158

6. Table 1 and 3 are maps?

7. Discussion is sound and well written. 

Author Response

Reviewer 2 :

Comments and Suggestions for Authors

First of all , authors used focused ion beam (FIB), which is indeed a state if the art technique and their results are quite impressive. 
Comment 1. Abstract is well written

Answer 1: Indeed, this technique allows for a wide range of characterization studies

Comment 2.  Regarding related studies only some references are provided RefS. 13-14. It is needed to provide literature corespond to previously conducted studies. 

Answer 2: The few related references are explained by the fact that there is very little usable literature on this specific case, hence the innovative positioning of this coating, which is the subject of an industrial invention patent recently filed and currently in force.

Comment 3. In introduction section some refrences are too old. Additionaly, gap of study is missing which can be tackle after dealing with point 2

Answer 3: As mentioned in question 2, the limited literature on this specific case does not allow us to provide recent and in-depth comparison elements. Additionally, given the novelty of our approach, we are unable to provide more relevant retrospective elements.
Comment 4. Experimental portion is well organized

Answer 4: Thank you
Comment 5. Rewrite sentences 157-158

Answer 5: It’s done

Comment 6. Table 1 and 3 are maps?

Answer 6: Yes,

Comment 7. Discussion is sound and well written. 

Answer 7: Thank you

Round 2

Reviewer 1 Report

Comments and Suggestions for Authors

The authors provided partial answers to the questions without clarifying the text itself, which makes it difficult for readers to follow their line of thinking. Only minor comments have been addressed, and the final PDF still includes the review track, hindering the evaluation of improvements. The authors frequently refer to a complementary study that is currently being published. I recommend resubmitting the paper once the other study is published.

Comments on the Quality of English Language

ok

Author Response

Thank you for your comments regarding our manuscript. We appreciate your feedback and are eager to improve our article for the benefit of the scientific community.

However, most of your comments on the major points to address seem to request clarification of your understanding of the manuscript rather than providing concrete and precise suggestions for improvement. Unlike the minor comments, which were specific suggestions that we have addressed according to your recommendations, the major points lack clear directives.Nevertheless, we have done our best to provide responses and hope we have met your expectations.

Regarding the revision marks in the manuscript file, this is a strict requirement from the journal, which mandates that authors leave these marks during modifications to track changes throughout the various corrections and improvements. This point is beyond our control and adheres to the journal's requirements.

In order to improve our work based on your constructive suggestions, we would be grateful if you could reformulate your comments on the major points with concrete and precise improvement suggestions. This will enable us to make more suitable and relevant modifications to our manuscript.